# Transforming Probabilistic Programs into Algebraic Circuits for Inference and Learning

**Pedro Zuidberg Dos Martires**
KU Leuven

**Vincent Derkinderen**
KU Leuven

**Robin Manhaeve**
KU Leuven

**Wannes Meert**
KU Leuven

**Angelika Kimmig**
Cardiff University

**Luc De Raedt**
KU Leuven

## Abstract

Probabilistic (logic) programs are routinely compiled into arithmetic circuits. During such a compilation step, the logic representation of a probabilistic program is transformed into an arithmetic representation. We show that this transformation and the resulting circuits cannot only be used for discrete probabilistic inference, but also for a number of other tasks such as differentiation, learning and probabilistic inference in the discrete-continuous domain.

## 1 Probabilistic Programming via Weighted Model Counting

Probabilistic programming extends high-level general purpose programming languages with probabilistic primitives. This allows for concisely modelling complex probability distributions without having to deal with the intricacies of probabilistic inference. One such probabilistic programming language is ProbLog (Fierens et al., 2015), where a probabilistic model consists of a set of probabilistic logical rules and probabilistic logical facts. We give an example program in Figure 1a. ProbLog performs probabilistic inference by reducing the task of computing the probability of a query being satisfied to the task of weighted model counting (WMC) (Chavira and Darwiche, 2008).

**Definition 1.** *(WMC) Given are 1) a propositional logic theory $\phi$ over a set of variables $\mathbf{b}$ representing the program and query and 2) a labeling function $\alpha : \mathcal{L} \to [0, 1]$, mapping literals in $\mathcal{L}$ from the variables in $\mathbf{b}$ to their probabilities. The* **weighted model count** *of a theory $\phi$ is then defined as:*

$$WMC(\phi, \alpha|\mathbf{b}) = \sum_{\mathbf{b}_\mathcal{I} \models \phi} \prod_{b_i \in \mathbf{b}_\mathcal{I}} \alpha(b_i) \tag{1}$$

where $\mathbf{b}_\mathcal{I}$ denotes an interpretation satisfying $\phi$.

```
1  0.6::burglary.
2  0.2::earthquake.
3  0.5::alarm_on.
4
5  alarm :- alarm_on, burglary.
6  alarm :- alarm_on, earthquake.
```

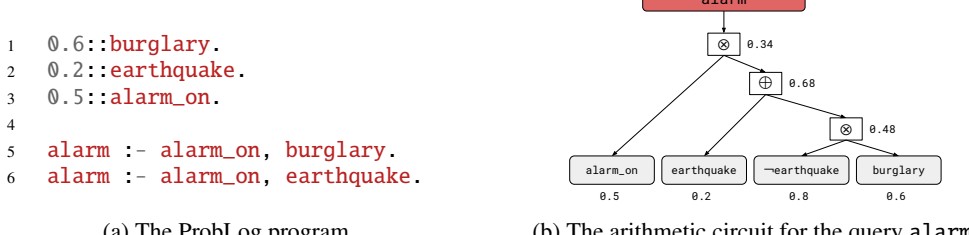

(a) The ProbLog program

(b) The arithmetic circuit for the query `alarm`.

Figure 1: An arithmetic circuit is used in ProbLog to answer probabilistic queries. Figure 1b shows the circuit used to compute the probability of the alarm going off (Figure 1a).

As WMC is in essence a counting problem, it falls into the #P-complete complexity class (Valiant, 1979). A popular technique in the (weighted) model counting literature to manage this complexity is

Program Transformations for Machine Learning (NeurIPS 2019 Workshop), Vancouver, Canada.

knowledge compilation (KC) (Darwiche and Marquis, 2002). The idea is to split up probabilistic inference into a computationally hard offline step, which compiles the source representation into a target representation (via knowledge compilation), which is then used for efficient query answering in a later cheap online step. A popular target representation is the language of sentential decision diagrams (SDD) (Darwiche, 2011), which are related to the well known binary decision diagrams (Bryant, 1986)[1].

The advantage of solving WMC via knowledge compilation, lies in the fact that the target representation can trivially be mapped to so-called arithmetic circuits (Darwiche and Marquis, 2002) — simply by replacing conjunctions and disjunctions with multiplications and additions, respectively. We give an example of such an arithmetic circuit in Figure 1b. Intuitively, ACs can be viewed as computation graphs to compute the WMC, cf. Equation 1, efficiently.

Going from the source code (a ProbLog program) to an arithmetic circuit (and the probability of the query) requires several program transformations, besides the knowledge compilation step. An overview of the different transformations performed by the ProbLog2 system (Dries et al., 2015) is shown in Figure 2. The first step consists of taking the weighted logic program and grounding it using a Prolog-based grounder. The obtained ground program — represented as a logical formula — is then compiled into a target representation (SDD or d-DNNF), on which weighted model counting can be performed in order to obtain the desired probability.

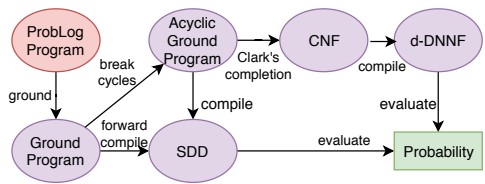

Figure 2: Overview of the primary program transformation steps in the ProbLog2 system.

Even though the content of this section and the remainder of the paper is presented in the context of probabilistic logics and probabilistic logic programming, the principles also apply to other programming languages, e.g. (Holtzen, Millstein, and Van den Broeck, 2019). We will focus on a logical perspective in the present note due to spatial constraints.

## 2 Algebraic Programming via Algebraic Model Counting

From an algebraic point of view, probabilistic inference boils down to performing computations in the (commutative) probability semiring $(\mathbb{R}_{\geq 0}, +, \times, 0, 1)$. Probabilistic inference can, hence, also be regarded as a specific case of a more general inference problem where multiplication and addition are generalised to the semiring operations $\otimes$ and $\oplus$, and their respective neutral elements $e^{\otimes}$ and $e^{\oplus}$. In reminiscence of model counting (also referred to as #SAT) (Gomes, Sabharwal, and Selman, 2008) and weighted model counting (Chavira and Darwiche, 2008), Kimmig, Van den Broeck, and De Raedt (2017) dub this generalized inference problem *algebraic model counting* and define it as follows.

**Definition 2.** *(Algebraic model counting) Given are 1) a propositional logic theory $\phi$ over a set of variables* **b***, 2) a commutative semiring $(\mathcal{A}, \oplus, \otimes, e^{\oplus}, e^{\otimes})$, 3) a labeling function $\alpha : \mathcal{L} \to \mathcal{A}$, mapping literals in $\mathcal{L}$ from the variables in* **b** *to values from the semiring set $\mathcal{A}$. The* **algebraic model count** *of a theory $\phi$ is then defined as: $AMC(\phi, \alpha | \mathbf{b}) = \bigoplus_{\mathbf{b}_I \models \phi} \bigotimes_{b_i \in \mathbf{b}_I} \alpha(b_i)$, where $\mathbf{b}_I$ denotes an interpretation satisfying $\phi$.*

This generalisation raises the question as to whether there are any other inference tasks that fit this more general inference problem. This question was answered affirmatively in (Kimmig, Van den Broeck, and De Raedt, 2011, 2017), where the authors show that a wide range of common AI problems can be formulated in the semiring setting (see Table 1 for a shortened list of examples). As the knowledge compilation approach described before only relies on semiring properties, it is also amenable to algebraic model counting — simply replace the multiplication and addition in the arithmetic circuit by $\otimes$ and $\oplus$ respectively. From now on, we will refer to circuits in the algebraic model counting context as *algebraic circuits*. Kimmig, Van den Broeck, and De Raedt also introduced the semantics for algebraic ProbLog (aProblog), which generalizes ProbLog (Fierens et al., 2015) to a semiring setting where the semiring is a parameter of the *algebraic program*. The ProbLog2

---

[1]Note that WMC can also be computed in an approximate fashion (Meel, 2018).

system has also been extended to allow users to define custom semirings in order to perform both probabilistic and algebraic programming.

| | $\mathcal{A}$ | $e^\oplus$ | $e^\otimes$ | $a \oplus b$ | $a \otimes b$ | $\alpha(b_i)$ | $\alpha(\neg b_i)$ |
|---|---|---|---|---|---|---|---|
| PROB | $\mathbb{R}_{\geq 0}$ | 0 | 1 | $a+b$ | $a \times b$ | $\alpha(b_i) \in [0,1]$ | $1-\alpha(b_i)$ |
| MPE | $\mathbb{R}_{\geq 0}$ | 0 | 1 | $\max(a,b)$ | $a \times b$ | $\alpha(b_i) \in [0,1]$ | $1-\alpha(b_i)$ |
| #SAT | $\mathbb{N}$ | 0 | 1 | $a+b$ | $a \times b$ | 1 | 1 |

Table 1: Examples of inference tasks common in artificial intelligence, modelled in the algebraic model counting setting (PROB=probability, MPE= most probable explanation). The operations + and × in the table denote the common addition and multiplication on the real and natural numbers respectively.

# 3 Transformations on algebraic circuits

In the context of probabilistic programming, algebraic circuits represent transformed programs, which can themselves further be transformed. In this section we briefly discuss five such transformations that have been studied in the algebraic circuits and probabilistic programming literature.

**Manipulating the structure** A first example of such a transformation involves changing the structure of the algebraic circuit. An advantage of using the SDD representation to perform such structural changes lies in the fact that conjunctions and disjunctions of two SDDs, as well as the negation of an SDD, can be computed in polytime (Darwiche, 2011). This can be used to compute the conditional probability of a query, i.e. the probability in presence of some evidence that has to hold. That is, one would like to compute the following $p(q|e) = \frac{p(q \wedge e)}{p(e)}$, where $q$ denotes the query and $e$ the evidence we condition on. This means if we represent $q$ and $e$ as SDDs the conditional probability can be computed in polytime.

**Relabeling the leaves** Suppose we are given a probabilistic program represented as an algebraic circuit with elements of a given semiring in the leaves. A straightforward program transformation to perform is the relabeling of the leaves in order to obtain a different program. Even though the (logical) structure of the program/algebraic circuit does not change under such a transformation it induces a new program. One can also think of the structure given by an arithmetic circuit as inducing a whole family of probabilistic programs. This transformation is essential when performing parameter learning, where the structure remains identical and the parameters are updated in an iterative fashion (Gutmann, Thon, and De Raedt, 2011).

**Transpilation to computation graphs** Probabilistic inference in algebraic circuits is carried out by performing a single bottom-up pass from the terminal nodes to the root. This bottom up pass can be mapped to a computation graph and algebraic circuits have already been successfully transpiled to computation graphs in TensorFlow (Abadi et al., 2015) in order to perform sampling based probabilistic inference for discrete-continuous problems (Zuidberg Dos Martires, Dries, and De Raedt, 2019). This direction of research has also been explored for sum-product networks (Poon and Domingos, 2011; Peharz et al., 2015) — a close relative of arithmetic circuits. Inference and learning algorithms for sum-product-networks can now easily be interfaced through Python libraries, which represent sum-product-networks again as TensorFlow computation graphs (Pronobis, Ranganath, and Rao, 2017; Molina et al., 2019). Since the irregular structure of ACs complicates a mapping to hardware (i.e., many GPU kernels), a dedicated accelerator has been proposed by Shah et al. (2019).

**Differentiation** A specific semiring, which we have not yet mentioned and which is specifically well suited for learning, is the gradient semiring (Eisner, 2002). Manhaeve et al. (2018) showed how to seamlessly integrate neural networks within probabilistic programming by using the gradient semiring and the concept of *neural predicates*. Orsini, Frasconi, and De Raedt (2017) related the gradient semiring to dual numbers and established thereby the link to forward mode automatic differentiation (Baydin et al., 2018). A related approach, however more akin to reverse mode automatic differentiation, was taken in (Dar-

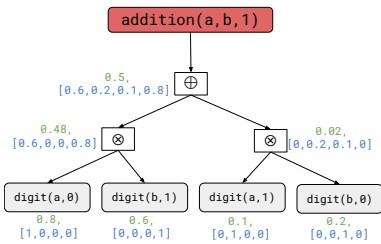

Figure 3: An AC that utilizes the gradient semiring. The number in green next to the node is the probability, and the numbers in blue below the gradient.

wiche, 2003). The proposed approach compiles a Bayesian network over discrete random variables into a *network polynomial*, which represents the algebraic circuit, and hence the underlying logical structure, as a differentiable structure. An initial exploration of this idea into the discrete-continuous domain is presented in (Zuidberg Dos Martires, 2019).

**Static circuit analysis** Static circuit analysis is a technique, introduced in (Kolb, Zuidberg Dos Martires, and De Raedt, 2019), employed in the domain of hybrid probabilistic inference, where the weighted model counting task is extended from the discrete domain to the continuous domain. This extension is referred to as weighted model integration (WMI) (Belle, Passerini, and Van den Broeck, 2015). In (Zuidberg Dos Martires, Dries, and De Raedt, 2019) it was shown that WMI, just as WMC, can be framed in the algebraic model counting setting. The key difference between WMI and ordinary AMC is the presence of an integration step for the WMI task. This intuitively implies that we need to perform an integration on the algebraic model count if we want to cast WMI using AMC: "WMI = ∫ AMC". More specifically, WMI can be solved by taking the integral of the symbolic structure provided as output by the algebraic circuit. This makes WMI amenable to powerful knowledge compilation technology and to a formulation as an aProbLog program (Zuidberg Dos Martires, Dries, and De Raedt, 2018).

Kolb, Zuidberg Dos Martires, and De Raedt (2019) showed that the presence of continuous random variables presents further opportunities for program transformations to speed up probabilistic inference in the hybrid domain. The idea is quite simple: once an algebraic circuit is obtained, a static analysis is performed by traversing the circuit once in a top-down fashion in linear time. The goal of this analysis is to figure out how deep into the circuit we can push the otherwise posterior integration (Figure 4). This transformation involves the analysis of the circuit to push additional operations into the desired place.

$$\int \left( [\![x>0]\!][\![x<1]\!][\![y<1]\!][\![x \leq y]\!] + \right.$$

$$\left. [\![x>0]\!][\![x<1]\!][\![y>1/2]\!][\![x>y]\!] \right) 2xy\,dx\,dy$$

$$= 2 \int_{(x>0)\wedge(x<1)} \left( \int_{(y<1)\wedge(x\leq y)} y\,dy + \int_{(y>1/2)\wedge(x>y)} y\,dy \right) x\,dx$$

(a) The AC encoding the inference problem.  (b) The resulting integration after transformation.

Figure 4: An inference problem in the discrete-continuous domain. The weight function which we want to integrate over is $2xy$. A static circuit analysis allows us to determine up to which point we can push the integration over $x$ and $y$ inside of the diagram. Instead of having an indefinite integration over indicator functions (in Iverson bracket notation (Knuth, 1992)), we have nested definite integrals.

## 4  Conclusions & Outlook

We have shown how different program transformations in the form of algebraic circuit transformations are equivalent to solving specific tasks in the field of artificial intelligence. The different transformations that can be performed on algebraic circuits have most often been studied and implemented in isolation of each other. Going forward, we would like to investigate if established techniques in compiler technologies and auto-differentiation can be applied to the transformations presented in this paper. This would constitute a step towards concisely tying together the different transformations that can be performed on algebraic circuits.

## Acknowledgements

This work is supported by the Research Foundation Flanders (FWO) under the Data-Driven Logistics project (S007318N), the CHIST-ERA and FWO under the H2020 project ReGround (G0D7215N), and the ERC AdG SYNTH(694980). RM is supported by the FWO (1S61718N).

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
