# OpenReview forum: "Transforming Probabilistic Programs into Algebraic Circuits for Inference and Learning"
_NeurIPS.cc/2019/Workshop/Program_Transformations — Program Transformations @NeurIPS2019 Poster_

### Official Review · AnonReviewer1 · 2019-09-26
**An interesting topic and seemingly technically sound execution**

**Confidence:** 3
**Rating:** 7

**Review:**

This paper deals with the transformation of probabilistic programs into logical circuits. Many of the specifics are out of my domain of speciality, but I found the topic interesting and relevant for the workshop, the writing clear, and found no glaring technical errors (again, this is not my field, so I could only check for what I know). It is a good submission, and we should accept it.

---

### Official Review · AnonReviewer2 · 2019-09-28
**Largely a survey of previous work/proposed future work, but an interesting synthesis and within the workshop's scope**

**Confidence:** 2
**Rating:** 7

**Review:**

This paper demonstrates how algebraic circuit transformations are equivalent to tasks such as differentiation and probabilistic inference. Much of the submission is a survey of previous work/agenda for future work on program transformations in algebraic circuits and PPLs, and it’s unclear to me whether there’s a substantial original research contribution. Nevertheless, I think that the paper contains an interesting perspective, is well-motivated, and within scope of the workshop.

---

### Decision · Program_Chairs · 2019-10-01

**Decision:**

Accept (Poster)

**Comment:**

The authors make interesting connections between AD/ML and algebraic circuits that we are eager to hear more about. The reviewers did mention that it is unclear in the abstract what are novel contributions and what is existing work.